# TXNIP Regulates Natural Killer Cell-Mediated Innate Immunity by Inhibiting IFN-γ Production during Bacterial Infection

**DOI:** 10.3390/ijms21249499

**Published:** 2020-12-14

**Authors:** Dong Oh Kim, Jae-Eun Byun, Won Sam Kim, Mi Jeong Kim, Jung Ha Choi, Hanna Kim, Eunji Choi, Tae-Don Kim, Suk Ran Yoon, Ji-Yoon Noh, Young-Jun Park, Jungwoon Lee, Hee Jun Cho, Hee Gu Lee, Sang-Hyun Min, Inpyo Choi, Haiyoung Jung

**Affiliations:** 1Department of Innovative Toxicology Research, Korea Institute of Toxicology, Yuseong-gu, Daejeon 34114, Korea; dongoh.kim@kitox.re.kr; 2Immunotherapy Research Center, Korea Research Institute of Bioscience and Biotechnology (KRIBB), Yuseong-gu, Daejeon 34141, Korea; quswodms@kribb.re.kr (J.-E.B.); kwasa@ensolbio.co.kr (W.S.K.); cjhsong5284@kribb.re.kr (J.H.C.); hanna@kribb.re.kr (H.K.); vnfmsgksmf89@kribb.re.kr (E.C.); tdkim@kribb.re.kr (T.-D.K.); sryoon@kribb.re.kr (S.R.Y.); nohj16@kribb.re.kr (J.-Y.N.); hjcho@kribb.re.kr (H.J.C.); hglee@kribb.re.kr (H.G.L.); 3Department of Biochemistry, School of Life Sciences, Chungbuk National University, Cheongju 28644, Korea; 4Research Center for Bioconvergence Analysis, Korea Basic Science Institute, Cheongju 28119, Korea; kmj0607@kbsi.re.kr; 5Department of Functional Genomics, Korea University of Science and Technology (UST), Yuseong-gu, Daejeon 34113, Korea; 6Environmental Disease Research Center, Korea Research Institute of Bioscience and Biotechnology (KRIBB), Yuseong-gu, Daejeon 34141, Korea; pyj71@kribb.re.kr (Y.-J.P.); jwlee821@kribb.re.kr (J.L.); 7Department of Biomolecular Science, Korea University of Science and Technology (UST), Yuseong-gu, Daejeon 34113, Korea; 8New Drug Development Center, Daegu-Gyeongbuk Medical Innovation Foundation (DGMIF), 80 Chumbokro Dong-gu, Daegu 41061, Korea; shmin03@dgmif.re.kr

**Keywords:** NK cell, IFN-γ, TXNIP, TAK1, toll-like receptor (TLR), bacterial infection

## Abstract

The function of natural killer (NK) cell-derived interferon-γ (IFN-γ) expands to remove pathogens by increasing the ability of innate immune cells. Here, we identified the critical role of thioredoxin-interacting protein (TXNIP) in the production of IFN-γ in NK cells during bacterial infection. TXNIP inhibited the production of IFN-γ and the activation of transforming growth factor β-activated kinase 1 (TAK1) activity in primary mouse and human NK cells. TXNIP directly interacted with TAK1 and inhibited TAK1 activity by interfering with the complex formation between TAK1 and TAK1 binding protein 1 (TAB1). *Txnip*^−/−^ (KO) NK cells enhanced the activation of macrophages by inducing IFN-γ production during Pam3CSK4 stimulation or Staphylococcus aureus (*S. aureus*) infection and contributed to expedite the bacterial clearance. Our findings suggest that NK cell-derived IFN-γ is critical for host defense and that TXNIP plays an important role as an inhibitor of NK cell-mediated macrophage activation by inhibiting the production of IFN-γ during bacterial infection.

## 1. Introduction

Natural killer (NK) cells serve as a crucial first line of defense against tumors and play an important role in innate immune responses in various tissues during the early steps of infection, inflammation, and tissue injury [1,2,3]. NK cells express pattern recognition receptors (PRRs), including toll-like receptors (TLRs), which are one of the most important PRRs and are expressed on most immune cells, such as dendritic cells (DCs), monocytes, macrophages, T cells, and B cells [4,5]. During the early phase of bacterial infection, NK cells mainly produce interferon-γ (IFN-γ), which can prime macrophage activation and contribute to potent innate immune responses, including the secretion of tumor necrosis factor α (TNF-α), interleukin 6 (IL-6) and interleukin 1β (IL-1β) [6,7]. Interestingly, recent reports have suggested that NK cells can be directly activated by numerous bacteria using their TLRs and consequently induce the production of cytokines such as IFN-γ, TNF-α, and granulocyte-macrophage colony-stimulating factor (GM-CSF) [8,9,10]. However, the nature of these bacteria-derived signaling pathways to produce cytokines and how they function in NK cells during bacterial infection remain poorly understood.

Pathogen-associated molecular patterns (PAMPs) are microbial molecules such as endotoxins and many other components. PAMP recognition by TLRs induces the activation of mitogen-activated protein kinases (MAPKs), nuclear factor-κB (NF-κB) and activator protein 1 (AP-1), and results in an effective immune response orchestrated by the production of pro- and anti-inflammatory cytokines [11,12,13]. Transforming growth factor beta-activated kinase 1 (TAK1) is a member of the mitogen-activated protein kinase kinase kinase (MAPKKK) family and is a critical mediator downstream of PAMP-TLR- NF-κB/AP-1 signaling pathways [14]. A TAK1 complex composed of TAK1 and the adapter proteins TAK1-binding protein 1 (TAB1) and TAB2 transduces the signal to c-Jun N-terminal kinase (JNK) and p38 via phosphorylation of IκB kinase (IKK), mitogen-activated protein kinase kinase 4/7 (MKK4/7), and MKK3/6 [15]. Ultimately, this signaling activates NF-κB, AP-1, and other transcription factors to induce the production of cytokines, which recruit and activate immune cells such as neutrophils and macrophages to eliminate microbes [16,17,18,19]. However, how the formation of the TAK1 complex is controlled or activated in NK cells during bacterial infection remains largely undefined.

Thioredoxin-interacting protein (TXNIP), initially discovered as a vitamin D3-induced gene in leukemia, was identified as a thioredoxin (TRX)-binding protein by yeast two-hybrid analyses [20,21]. TXNIP has diverse functions involved in many cellular processes, including the regulation of differentiation, cell cycle, tumor, aging, metabolism, and inflammation [22]. TXNIP expression is upregulated by various stresses, such as H_2_O_2_ exposure, UV irradiation, heat shock, serum deprivation, and transforming growth factor-β (TGF-β) stimulation [23,24]. TXNIP is significantly downregulated in a variety of tumors, including breast, renal, and gastro-intestinal cancers [25,26]. According to our previous reports, TXNIP is involved in the development and function of NK cells [27] and linked NO synthesis and NLRP3 inflammasome activation during endotoxic shock [28]. The loss of *Txnip* in hematopoietic stem cells causes severe damage under stress conditions [29,30].

Here, we investigate the function of TXNIP in the production of IFN-γ in NK cells to activate macrophages during bacterial infection in vitro and in vivo. We present that TXNIP acts as a critical inhibitor of IFN-γ production by inhibiting TAK1 activity via direct interaction in NK cells. We also prove that the loss of *Txnip* in NK cells increases their sensitivity to activation, stimulating the production of larger amounts of IFN-γ in *Txnip*^−/−^ (KO) NK cells than in wild-type (WT) NK cells under Pam3CSK4, a TLR2/TLR1 agonist, treatment and bacterial infection conditions. These findings suggest that TXNIP may be a new critical regulator and a therapeutic target in NK cell-mediated innate immune responses during bacterial infections.

## 2. Results

### 2.1. Differential Induction of IFN-γ by TLR Agonists in KO NK Cells

According to our previous report, TXNIP controls the development of NK cells [27]. However, whether TXNIP can regulate other functions of NK cells, including cytotoxicity and/or cytokine production, especially under PAMP stimulation or bacterial infection, has not been thoroughly investigated. IFN-γ is mainly produced by NK cells and is one of the most important cytokines for the activation of granulocytes, which release proinflammatory cytokines such as IL-6 and IL-1β under bacterial infection [4,31,32,33]. In our previous study, the loss of *Txnip* in innate immune cells, such as neutrophils and macrophages, did not result in any differences in the production of proinflammatory cytokines under direct stimulation with PAMPs or bacterial infection in vitro; however, even though KO mice had fewer NK cells, compared with WT mice, KO mice showed similar production levels of proinflammatory cytokines and were hypersusceptible to endotoxic shock [28]. Based on these results, we hypothesized that KO NK cells might produce more IFN-γ to activate innate immune cells than WT NK cells under bacterial infection. To investigate whether KO NK cells produce more IFN-γ under bacterial infection than WT NK cells, we obtained highly purified NK cells from the spleens of WT and KO mice. These NK cells were treated with various PAMPs, microbial molecules, such as lipoteichoic acid (LTA, a TLR2/TLR6 agonist), Pam3CSK4 (a TLR1/TLR2 agonist), Poly (I:C) (a TLR3 agonist), and lipopolysaccharide (LPS, a TLR4 agonist). All TLR agonists induced the secretion of IFN-γ in both WT and KO NK cells, while Pam3CSK4 and LPS differentially induced the secretion of IFN-γ in KO NK cells (Figure 1A). Based on these results, we selected the TLR1/TLR2 signaling pathway among them to prove the regulatory functions of TXNIP in the production of IFN-γ in NK cells. Pam3CSK4 (Pam) induced the accumulation of secreted IFN-γ in both WT and KO NK cells in a time-dependent manner (Figure 1B), and the production of IFN-γ was determined as the percent of IFN-γ expressing cells by flow cytometric analysis through intracellular staining (Figure 1C,D). However, Pam treatment could not induce the production of TNF-α or perforin in both WT and KO NK cells (Appendix A). Interestingly, the cytotoxicity of NK cells against YAC-1 cells and the expression of activating receptors or inhibitory receptors were not significantly regulated by Pam treatment in WT and KO NK cells (Figure 1E,F). These data indicate that TXNIP inhibits the production of IFN-γ in NK cells but not the cytotoxic activity of NK cells during Pam stimulation.

### 2.2. The Loss of Txnip Activates TAK1 and Induces IFN-γ Production in NK Cells

To investigate how TXNIP regulated the production of IFN-γ, we explored the expression of TLR1 and 2 and the phosphorylation of their downstream kinases in both WT and KO NK cells during Pam stimulation in vitro. As shown in Figure 2A, the expression of TLR1 or TLR2 was not significantly changed on the cell surface between WT and KO NK cells at 12 h after Pam treatment, and TLR2 mRNA was not induced (Appendix A). Next, to determine the activation of the TLR1/2 signaling pathway in NK cells by Pam treatment, we examined the phosphorylation levels of signaling molecules belonging to the TLR1/2 signaling pathway in NK cells during Pam stimulation by western blotting. The expression of TLR2 and the phosphorylation of IRAK1 were similar between WT and KO NK cells, but interestingly, TAK1 and its downstream kinases (ERK, JNK, and p38) were more highly phosphorylated in KO NK cells than in WT NK cells, and TXNIP showed the opposite response to Pam stimulation (Figure 2B and Appendix A). The mRNA expression of TXNIP was also regulated in a manner opposite the regulation of the expression of IFN-γ by Pam stimulation in NK cells (Appendix A). Additionally, to evaluate the contribution of TLR1/2 in the production of IFN-γ by Pam stimulation, we treated NK cells with Pam and CU-CPT22, a TLR1/2-specific antagonist. The production of IFN-γ was markedly inhibited by CU-CPT22 in both WT and KO NK cells (Figure 2C). Next, to investigate the function of TXNIP in primary human NK cells, we prepared CD3^−^/CD56^+^ NK cells from umbilical cord blood (CB) and transduced *Txnip* siRNA into the cells. Transient knockdown of *Txnip* (Figure 2D) significantly induced the production of IFN-γ (Figure 2E) and the phosphorylation of TAK1 (Figure 2F,G) in primary human NK cells during Pam stimulation. Based on these results, we identified a negative correlation between TXNIP and IFN-γ production and found that TAK1 might be a putative target of TXNIP in the regulation of IFN-γ production in NK cells during Pam stimulation.

### 2.3. Regulation of TAK1 Activity by TXNIP via Physical Interaction

To investigate the regulatory mechanism of TXNIP on TAK1 activation, we performed glutathione S-transferase (GST) pull-down assays. Cotransfected TAK1 was precipitated with TXNIP in HEK 293 cells (Figure 3A). The direct interaction between TXNIP and TAK1 was further confirmed using purified recombinant TXNIP and TAK1 proteins (Appendix A). Next, to examine whether TRL1/2 signaling regulated the interaction between TXNIP and TAK1, we performed an immunoprecipitation assay in human NK92 cells and a proximity ligation assay (PLA) in primary mouse NK cells. Interestingly, the interaction between TXNIP and TAK1 was decreased by Pam treatment in NK92 cells (Figure 3B) and primary mouse NK cells (Figure 3C). To determine the possible interaction region for this interaction, we generated two deletion mutants of TXNIP, 1–149 and 150C [30], and four deletion mutants of TAK1, 1–34, 35–291, 292–478, and 479–579. GST pull-down assays revealed that TAK1 interacted with the 1–149 region of TXNIP (Figure 3D) and that TXNIP interacted with the 35–291 region of TAK1 (Figure 3E). To identify whether TAK1 kinase activity was needed for TXNIP binding, we performed a GST pull-down assay between TXNIP and TAK1 or TXNIP and TAK1 (K63W), a kinase dead mutant. Cotransfection of TAK1 (K63W) with TXNIP showed a similar interaction with TAK1 (Appendix A). These results strongly suggest that TXNIP might regulate TAK1 activity as an upstream molecule. As shown in Figure 3E, TXNIP interacted with the 35–291 region of TAK1, which contained the TAB1 binding region [34,35]. Thus, we investigated whether TXNIP was involved in the interaction between TAK1 and TAB1, TAB2, or TAB3. As expected, TXNIP interfered with the interaction between TAK1 and TAB1 (Figure 3F) but not TAB2 or TAB3 (Appendix A). To further confirm the inhibitory effect of TXNIP on the interaction between TAK1 and TAB1, we examined the interaction between TAK1 and TAB1 following Pam treatment in primary NK cells using in situ PLA assays. The interaction between TAK1 and TAB1 was increased by Pam treatment, and the loss of *Txnip* markedly increased their interaction in NK cells (Appendix A). Recombinant TXNIP also inhibited TAK1 kinase activity in vitro in a dose-dependent manner (Figure 3G). These results suggested that TXNIP directly interacted with TAK1 and inhibited TAK1 kinase activity by blocking a complex between TAK1 and TAB1.

### 2.4. TXNIP Inhibits Pam3CSK4-Induced AP-1 Transcriptional Activity

The AP-1 heterodimer, Fos–Jun, cooperatively binds a composite DNA site of promoters and is one of the major transcription factors for IFN-γ expression [16,36]. To examine the impact of TXNIP on TAK1-mediated AP-1 transcriptional activity, we transfected increasing amounts of TXNIP, TXNIP (1–149), or TXNIP (150C) along with an AP-1 luciferase reporter and Pam treatment or TAK1. TXNIP significantly inhibited Pam3- and TAK1-mediated AP-1 transcriptional activity, and the nonbinding region of TXNIP, 150C, to TAK1 could not inhibit TAK1-mediated AP-1 transcriptional activity (Figure 4A,B). In addition, the activation of c-Fos was determined by phosphorylation using western blotting and immunostaining in WT and KO NK cells. The loss of *Txnip* markedly induced the activation of c-Fos in NK cells (Figure 4C,D). Next, to reveal the function of TAK1 kinase activity on the production of IFN-γ in NK cells, we incubated NK cells with 5Z-7-Oxozeaenol (5Z), a TAK1 inhibitor. As expected, 5Z efficiently inhibited the phosphorylation of TAK1 by Pam stimulation in both WT and KO NK cells (Figure 4E) and suppressed the phosphorylation of c-Jun (Figure 4F,G). Consequently, the inhibition of TAK1 activity resulted in a reduction in IFN-γ production in NK cells (Figure 4H). These results suggested that TXNIP antagonized Pam- or TAK1-mediated AP-1 transcriptional activity by inhibiting TAK1 activity via direct interaction and revealed that TAK1 activity was crucial for Pam-induced IFN-γ production in NK cells.

### 2.5. NK Cell-Dependent Activation of Macrophages In Vitro and In Vivo

According to recent studies, crosstalk occurs between NK cells and macrophages, which plays an important role in anti-infection responses, and macrophages are activated by IFN-γ, which is mainly produced by NK cells and CD4^+^ T cells via increasing receptors such as CD80, CD86 and MHC II [6,33,37,38]. To investigate whether NK cells activated macrophages in vitro, we prepared purified splenic NK cells from WT and KO mice and treated them with IL-2 (a positive control) or Pam with or without 5Z for 8 h. Then, phosphate-buffered saline (PBS)-washed NK cells were cocultured with RAW 264.7 cells, a murine macrophage cell line, for 16 h at a ratio of 1:5. Both Pam-stimulated WT and KO NK cells activated RAW 264.7 cells, and KO NK cells markedly induced the expression of CD80, CD86 and MHC II (Figure 5A–C) and the production of proinflammatory cytokines, including IL-1β, IL-6 and TNF-α in RAW 264.7 cells (Figure 5D–F). However, IL-2 alone could not activate RAW 264.7 cells, and 5Z dramatically inhibited Pam-mediated activation of RAW 264.7 cells (Figure 5A–F). KO NK cells also increased the frequency of F4/80^+^/CD11b^+^ RAW264.7 cells than those of WT NK cells (Appendix A). Next, to evaluate the impact of NK cells on macrophage activation in vivo, we depleted NK cells from WT mice using an α-NK1.1 (PK136) antibody (Ab) (Appendix A). Surprisingly, NK cell-depleted mice showed a significant reduction in Pam-induced immune responses in overall experiments. NK cell-depleted mice were intraperitoneally (i.p.) injected with Pam (Figure 5G); then, the production of IFN-γ (Figure 5H), the percentage of macrophages in the spleen (Figure 5I), the expression of CD80 and CD86 (Figure 5J,K), and the production of IL-1β and IL-6 (Figure 5L,M) were determined. The surface expression of CD80 and CD86 on macrophages (Appendix A) showed similar patterns with IL-1β and IL-6 secreted from macrophages in serum (Appendix A). The frequency of splenic NK cells, T cells, and macrophages also fluctuated in response to Pam treatment (Appendix A). In summary, we found the inhibitory function of TXNIP in NK cell-mediated activation of macrophages during Pam stimulation and revealed the impact of NK cells on macrophage activation in vivo.

### 2.6. Differential Activation of Macrophages by KO NK Cells In Vivo

To investigate whether KO NK cells could effectively impact macrophage activation in the early innate immune response in vivo and to prove the autonomous function of TXNIP in NK cells, we performed an experiment using adoptive-transfer mouse models. *Rag2*^−/−^ × *Il2rg*^−/−^ mice show decreased thymus, a reduction of the cellularity of other lymphoid organs as spleens and lymph nodes, and the complete absence of B, T, NK cells [39]. We transferred WT or KO NK cells into the tail vein of *Rag2*^−/−^ × *Il2rg*^−/−^ mice and then injected Pam i.p. at 3 days after the adoptive transfer (Figure 6A). WT or KO NK cells were similarly transferred to the spleens of recipient *Rag2*^−/−^ × *Il2rg*^−/−^ mice and maintained similar frequencies after Pam3 injection (Appendix A). As expected, mice that received KO NK cells showed significantly higher levels of IFN-γ (Figure 6B), an increased frequency of macrophages (Figure 6C,D), higher levels of CD80 and CD86 (Figure 6E,F), and increased levels of proinflammatory cytokines (Figure 6G,H). Next, to identify the contribution of NK cell-derived IFN-γ to macrophage activation in vivo, we also injected recipient mice with α-IFN-γ Ab to neutralize the function of IFN-γ (Figure 6A). The frequency of macrophages and the levels of IL-1β and IL-6 were significantly decreased by α-IFN-γ Ab injection in both WT and KO NK cell-recipient mice (Figure 6I–K). These data suggested that TXNIP played an important role in the production of IFN-γ in NK cells to activate macrophages during Pam stimulation in vivo.

### 2.7. KO NK Cells Induce a More Effective Bacterial Clearance In Vivo

As shown in Figure 6, KO NK cells induced higher levels of activation of macrophages than did WT NK cells in response to Pam stimulation in vivo. Next, we examined the regulation of host defense against bacterial infection by WT or KO NK cells. We prepared adoptive-transfer mouse models as previously shown in Figure 6 and then applied *S.*
*aureus*, a gram-positive bacterium, to the mouse models (Figure 7A). KO NK cell-recipient mice were resistant to *S. aureus*, resulting in significantly decreased bacterial growth in the kidneys (Figure 7B) and blood (Figure 7C). We also analyzed the cell phenotypes and cytokines in individual spleen and serum samples from mice that received WT or KO NK cells. The frequency of adoptive-transferred WT and KO NK cells was similar in recipient mice at 0 h and 12 h after *S.*
*aureus* infection (Appendix A). However, the frequency of splenic macrophages in KO NK cell-recipient mice was higher than that in WT NK cell-recipient mice (Figure 7D,E). The levels of IL-1β and IL-6 were also higher in KO NK cell-recipient mice than in WT NK cell-recipient mice (Figure 7F,G). IFN-γ levels in the serum were not detectable due to downregulation at late time points (data not shown). Thus, our results provided evidence that the regulation of INF-γ production by TXNIP in NK cells was crucial for the host defense against bacterial infection and revealed that NK cells played important roles in innate immunity by directly responding to bacterial infection via TLRs/TAK1/IFN-γ signaling pathways.

## 3. Discussion

Here, we showed that the loss of *Txnip* in NK cells resulted in significantly increased IFN-γ production and activation of macrophages during Pam stimulation. IFN-γ is a critical, multipotent cytokine for innate and adaptive immunity against viral and intracellular bacterial infections. Additionally, IFN-γ contributes to macrophage activation by increasing phagocytosis and priming the production of proinflammatory cytokines [8,40]. NK cells are crucial producers of IFN-γ in the early immune response that contributes to the pathogen clearance process and induces the inflammatory response [41,42]. NK cells express TLRs, allowing them to respond to PAMPs, and the activation of TLRs by their agonists or bacterial infections induces the production of IFN-γ in NK cells [8,43]. Under certain conditions, such as culture or stimulation, intracellular TLR2 and TLR4 traffic and move to the cell surface [44,45,46]. We also showed the expression of TLR1 and TLR2 on the surface and identified the induction of IFN-γ in NK cells by various TLR agonist treatments.

TAK1 is known to participate in TLR-induced cellular signaling pathways to produce IFN-γ in NK cells [47,48]. TAK1 requires a protein activator, TAB1, to induce its full activation [49]. Ectopic expression of TAB1, together with TAK1, induces autophosphorylation of TAK1, thereby activating TAK1 kinase in vitro [50,51]. We identified the critical function of TAK1 activity in the production of IFN-γ in WT and KO NK cells. As shown in our results, TAK1 inhibitor, 5Z, significantly reduced the production of IFN-γ in NK cells during Pam stimulation. We found that TXNIP interacted with TAK1 and inhibited the kinase activity of TAK1 via interfering with the interaction between TAK1 and TAB1. We also investigated the regulation of TAK1 activity by TXNIP in primary human NK cells. We transduced *Txnip* siRNA into CD3^−^/CD56^+^ NK cells from CB. Transient knockdown of *Txnip* significantly induced the production of IFN-γ and the phosphorylation of TAK1 in primary human NK cells during Pam stimulation.

TXNIP is a multifunctional protein involved in a variety of cellular processes, including the regulation of differentiation, the cell cycle, cancer, cell aging, metabolism, inflammation, and immune cell regulation [22,30,52]. Dendritic cells derived from *Txnip*-deficient mice exhibit defects in regulating T cell activation and proliferation [53]. According to our previous report, TXNIP plays an important role in the development and function of NK cells [27] and mediates the context-dependent functions of TXNIP in different cells by interacting with signaling molecules, including p53, p38, macrophage migration inhibitory factor (MIF) and AKT [29,30,52,54]. As previously reported [27], because KO mice show defects in the development of NK cells, we adoptively transferred WT and KO NK cells into *Rag2*^−/−^ × *Il2rg*^−/−^ mice to evaluate the impact of NK cells during bacterial infection. Adoptive transfer of NK cells into *Rag2*^−/−^ × *Il2rg*^−/−^ mice revealed that KO NK cells enhanced the activation of macrophages by inducing IFN-γ production during Pam stimulation or *S. aureus* infection and contributed to expedite the bacterial clearance. In our previous report [27], we could not define distinct subsets of NK cells in functional maturation, because we did not have common markers for functional NK cell maturation at that time. Recent reports have suggested that functional NK cell maturation can be defined by the differential surface expression of CD27 and CD11b. To investigate the NK cell subsets in WT and KO mice, we tried to analyze frequency of CD27 and CD11b expression in CD3^−^NK1.1^+^ gated WT and KO NK cells. Interestingly, there was no effect on functional NK cell maturation by TXNIP deletion, but rather the final stage (CD27^−^CD11b^+^) maturation tended to increase in KO NK cells (Unpublished data). From these results, we could hypothesize that TXNIP may contribute to the immature stages of NK cell differentiation but not to the functional NK cell maturation. To understand or explain for roles of TXNIP in functional NK cell maturation, we need to do more detailed experiments. We expect to get answers through future study. From the series of experimental results, we could demonstrate that the loss of *Txnip* in NK cells significantly induced TAK1 activity, IFN-γ production, macrophage activation, and ultimately protected the host against bacterial infection.

In conclusion, NK cell-derived IFN-γ production is crucial for macrophage activation, and TXNIP plays a critical role in the production of IFN-γ in NK cells during bacterial infection. TXNIP directly inhibits TAK1 activity, which is an important signaling mediator of TLR1/2, by interfering with the interaction between TAK1 and TAB1. The loss of *Txnip* significantly enhances the production of IFN-γ in NK cells and the activation of macrophages, consequently removing bacteria during bacterial infection. These data suggest that TXNIP may be a potential therapeutic target for bacterial infectious diseases.

## 4. Materials and Methods

### 4.1. Mice

*Txnip*^−/−^ mice (C57BL/6) were generated as described previously (Lee et al., 2005). WT C57BL/6J mice and *Rag*2^−/−^ × *Il2rg*^−/−^ mice were purchased from Jackson Laboratory. Male mice were used for overall study. All mice were maintained under specific pathogen free (SPF) conditions and used at 7–10 weeks-age. All animal experiments were approved by the Institutional Animal Use and Care Committee of the Korea Research Institute of Bioscience and Biotechnology (KRIBB-AEC-18036) and were performed in accordance with the Guide for the Care and Use of Laboratory Animals published by the US National Institutes of Health.

### 4.2. Cell Culture and NK Cells Preparation

The murine macrophage-like cell line RAW264.7 and the murine lymphoma cell line YAC-1 cells were cultured in RPMI 1640 (WelGENE) supplemented with 10% FBS (Hyclone) and 1% penicillin streptomycin-amphotericin B (WelGENE) (referred to as complete RPMI medium). The human NK cell line, NK92 was cultured in alpha-MEM (Gibco) supplemented with 20% heat-inactivated FBS (Hyclone) and 2 mM L-glutamine. Human embryonic kidney (HEK) 293 cells were maintained in Dulbecco modified Eagle medium (DMEM) containing 10% FBS and 1% penicillin-streptomycin amphotericin B (referred to as complete DMEM medium). Primary mouse NK cells were isolated from the spleen using the mouse NK Cell Isolation Kit II (Miltenyi Biotec) and sorted using FACSFusion cell sorter (Becton Dickinson, San Jose, CA, USA) according to the manufacturer’s protocol. Enriched mouse NK cells (>90% NK1.1^+^) were cultured in RPMI 1640 (WelGENE) containing 10% fetal bovine serum (FBS) (Hyclone) and 1% penicillin-streptomycin-amphotericin B (WelGENE) supplemented with 30 ng/mL of recombinant mouse IL-15 (PeproTech) for all experiments in this paper. Primary human CD3^−^CD56^+^ NK cells were isolated from umbilical cord blood (CB) mononuclear cells (approved by Public Institutional Review Board designated by Ministry of Health and Welfare, P01-201610-31-002) using RosetteSep (Stem Cell Technologies, Vancouver, BC, Canada)—which depletes cluster of differentiation CD3^+^ T cells and red blood cells—followed by CD56 magnetic beads (Miltenyi Biotec, Bergisch Gladbach, Rhine-Westphalia, Germany). The cells were cultured in α-Minimal Essential Medium (Welgene, Gyeongsan, Korea) with IL-15 (100 U/mL), IL-21 (100 U/mL), and 1 µM of hydrocortisone (HC; Stem Cell Technologies, Vancouver, BC, Canada). Cells were maintained under 37 °C and 5% CO_2_ conditions.

### 4.3. Reagents and Antibodies (Abs)

Lipoteichoic acid from *Staphylococcus aureus* (LTA), Lipopolysaccaride from Escherichia coli, 055:B5 (LPS), CU CPT22 (Sigma, TLR2-TLR1 receptor inhibitor), and 5Z-7-Oxozeaenol (TAK1 inhibitor) were purchased from Sigma (St. Louis, MO, USA). Pam3CSK4 (Pam) and polyinosinic:polycytidylic acid [poly(I:C)] were purchased from Invivogen (San Diego, CA, USA). LPS, LTA, and Pam3CSK4 were used at a final concentration of 1 µg/mL and poly (I:C) was used at a final concentration of 50 µg/mL. And α-NK1.1 for NK cell depletion and α-IFN-γ for IFN-γ neutralization were purchased from Bio X Cell.

Antibodies used for immunofluorescence staining were anti–NK-1.1 (clone PK136) used as APC or fluorescein isothiocyanate (FITC) conjugates; anti–CD3 (clone 17A2) used as FITC or APC-Cy7; anti-IFN-γ (clone XMG1.2) used as a phycoerythrin (PE) or APC or PE-Cy7 conjugate; anti–Mac-1 (clone M1/70) used as PE-Cy7 conjugate; anti–Gr-1 (clone RB6-8C5) used as Alexa fluor 488 conjugates; anti-F4/80 (clone BM8) used as eFluor 506 conjugate; anti-CD80 (clone 16-10A1) used as PE conjugate; anti–CD86 (clone GL-1) used as Brilliant Violet 421 (BV421) conjugate; anti-Ly49D (clone 4E5) used as an FITC conjugate; and anti-CD314 (clone A10) used as a PE conjugate; anti–CD62L (clone MEL-14) used as FITC conjugates; anti–Ly49A (clone A1) used as PE conjugate; anti-Ly49C/I (clone 5E6) used as a PE conjugate; anti–Ly49G2 (clone 4D11) used as APC conjugate; anti–CD282 (clone T2.5) used as FITC or PE or Alexa Fluor 647 conjugates; anti-phospho-c-Jun (Ser73) (clone D47G9) used as PE conjugate; anti-I-A/I-E (clone M5/114.15.2) used as PE or APC conjugate. The cells were first incubated with anti-mouse immunoglobulin FcRg antibody (clone 2.4G2) before the addition of the appropriate combination of the above-mentioned fluorescence-conjugated antibodies against cell surface markers of interest. Intracellular IFN-γ and Perforin staining were performed according to the manufacturer’s instructions (BD). All stained cells were analyzed using FACSCanto II (Becton Dickinson, San Jose, CA, USA)

### 4.4. Flow Cytometry and Intracellular Staining

Cells were washed with FACS buffer (PBS containing 1% FCS and 2 mM EDTA) and labeled for 20 min at 4 °C with above-mentioned mAbs. When necessary, cells were incubated with anti-CD16/CD32 blocking Ab (eBioscience; 2.4G2). To analyze intracellular cytokine levels, cells were fixed and permeabilized with the cytofix/cytoperm kit, according to the manufacturer’s protocol (BD Biosciences). IFN-γ– and Perforin–producing cells were gated on CD3^−^NK1.1^+^ cells prior to analysis. Expression levels were acquired using FACSCanto II (BD Biosciences), and data were analyzed using DIVA software (BD Biosciences).

### 4.5. Plasmid Construction

FLAG-TAB1 and -TAB2 clones were kindly gifted by Professor Ki-Young Lee (Sungkyunkwan University School of Medicine, Korea). We constructed FLAG-tagged TXNIP (full), TXNIP (1–149), TXNIP (150C), TAK1 Kinase Dead form (K63W) and TAB3, and GST-fused TAK1, TAK1 (1–34), TAK1 (35–291), TAK1 (292–478), and TAK1 (479–579). Human TAK1 (hMU001673) and TAB3 (hMU005123) clones were provided from Korea Human Gene Bank, Medical Genomics Research center, KRIBB, Korea. We used following primers: for TXNIP (1–149), forward (EcoR1) 5′-GCG AAT TCA TGG TGA TGT TCA AGA AGA TC-3′, reverse (Sal1) 5′-GCG TCG ACT CAC ACC AGA TCC ACT ACT TCA AA-3′; for TXNIP (150C), forward (EcoR1) 5′-GCG AAT TCG ATG TCA ATA CCC CTG ATT TA-3′, reverse (Sal1) 5′-GCG TCG ACT CAC TGC ACA TTG TTG TTG AG-3′; for TAK1 full construction, forward (EcoR1) 5′-GCGAATTCATGTCTACAGCCTCTGCCGCC-3′, reverse (BamH1) 5′-GCGGATCCTCATGAAGTGCCTTGTCGTTT-3′; for TAK1 Kinase Dead form (K63W) construction, forward (EcoR1) 5′-GATGTTGCTATTTGGCAAATAGAAAGT-3′, reverse (BamH1) 5′-ACTTTCTATTTGCCAAATAGCAACATC-3′. For GST-fused TAK1 deletion mutants, we used the primers as follows: for TAK1 (1–34), forward (BamH1) 5′-GCG GAT CCA TGT CTA CAG CCT CTG CCG CC-3′, reverse (Not1) 5′-GCG CGG CCG CTC ACT CCT TGT AGT CGA TCT-3′; for TAK1 (35–291), forward (BamH1) 5′-GCG GAT CCT AGC TCC ACC TTC TCC AAC AAC-3′, reverse (Not1) 5′-GCG CGG CCG CTC AAA AGT ACC GCA TCA AGT-3′; for TAK1 (292–478), forward (BamH1) 5′-GCG GAT CCC CAG GAG CAG ATG AGC CAT TAC-3′, reverse (Not1) 5′-GCG CGG CCG CTC AGT GAT CCA GTG TAA GAT-3′; for TAK1 (479–579), forward (BamH1) 5′-GCG GAT CCC AAC TAC AGC CTC TAG CAC CGT-3′, reverse (Not1) 5′-GCG CGG CCG CTC ATG AAG TGC CTT GTC GTT-3′. For TAB3 Full construction, we used the primers as follows: for TAB3 (Full), forward (EcoR1) 5′-GCGAATTCATGGCGCAAAGCAGCC-3′, reverse (BamH1) 5′-GCGGATCCTCAGGTGTACCGTGGC-3′. All Primers were synthesized by Bioneer (Daejeon, Korea).

### 4.6. Transient Transfection 

Transient transfections of HEK 293 cells were performed using Lipofectamine and Plus Reagent (Invitrogen, Carlsbad, CA, USA) according to the manufacturer’s protocol. The cells were cotransfected with pEBG (GST) vector, pEBG-TXNIP, pEBG-TAK1, pEBG-TAK1 (1–34), pEBG-TAK1 (35–291), pEBG-TAK1 (292–478), pEBG-TAK1 (479–579), pEBG-TAK1 (K63W; Kinase dead form), pFLAG vector, pFLAG-TAK1, pFLAG-TXNIP, pFLAG-TXNIP (1–149), pFLAG-TXNIP (150C), pFLAG-TAB1, pFLAG-TAB2, pFLAG-TAB3 and pHA-TXNIP using Lipofectamine/Plus Reagent. For siRNA transfection, human CD3^−^/CD56^+^ NK cells were cultured in 6-well plate at a density of 3 × 10^6^ cells/mL for 12 h before transfection. Transfection (2 × 10^6^ cells/condition) was carried out using a NEPA21 electroporator (Nepagene, Chiba, Japan) with 300 nmol/100 µL of control or *Txnip* siRNA. TXNIP siRNA and control siRNA were acquired from Santa Cruz Biotechnology (SC-44943).The transfected cells were cultured for 24 h and then were treated with Pam3CSK4 (1 µg/mL) for 0.5 h (for confocal imaging) or 18 h (for ELISA assay).

### 4.7. Immunoprecipitation and Western Blotting

HEK 293 cells were used for an immunoprecipitation and highly purified sorted NK cells (99.5–99.9% NK1.1^+^CD3^−^) were used for a Western Blot. The cells were lysed with RIPA buffer (50 mM Tris–HCl, pH 7.5, 1% Nonidet P-40, 1 mM EDTA, 1 mM phenylmethylsulfonyl fluoride, l g/mL leupeptin, 1 mM sodium vanadate, and 150 mM NaCl). Cell lysates were incubated with glutathione Sepharose 4B (GE Healthcare, Uppsala, Sweden) for 3 h at 4 °C. Then it washed four times with lysis buffer. The combined proteins were suspended in SDS sample buffer, and then resolved on an 8–12% SDS-PAGE gels and transferred to PVDF membrane (Millipore, Bedford, MA, USA). Immunoprecipitation of endogenous TAK1 with endogenous TXNIP was analyzed in NK-92 cells. At 1 h after Pam3CSK4 (1 μg/mL) treatment, the cells were harvested and lysed in RIPA buffer and 1× protease inhibitor mixture (Calbiochem, San Diego, CA, USA). The cell lysates were then incubated with different Abs for 2–3 h at 4 °C. The Ag-Ab complexes were precipitated by incubation at 4 °C for 3 h with protein G-conjugated agarose (Roche).

The membrane was probed with primary antibodies specific to the following molecules: p-IRAK1, IRAK1, p-TAK1, TAK1, p-ERK, ERK, p-JNK, JNK, p-p38, p38, p-c-Fos, c-Fos, p-c-Jun, c-Jun, TXNIP (Cell Signaling), FLAG (Sigma), HA, GST, TLR2 and β-Actin (Santa Cruz). After incubation with peroxidase-conjugated anti-rabbit or anti-mouse IgG (Jackson ImmunoResearch), the signals were detected using SuperSignal West Pico Chemiluminescent Substrate (Pierce). Western blots were imaged using WSE-6100 LuminoGraph (ATTO, Tokyo, Japan).

### 4.8. In Situ Proximity Ligation Assay (PLA) and Confocal Imaging

Freshly isolated NK cells were plated on fibronectin-coated glass coverslips (Neuvitro) at 4 °C for 10–20 min and treated with Pam3CSK4 in NK cells media for 30 min. After stimulation, the cells were washed with PBS and fixed for 20 min at room temperature with 4% paraformaldehyde in PBS, followed by permeabilization with 0.2% Triton X-100 for 15 min at 4 °C. For in situ PLA assay of the interaction between TAK1 and TXNIP or TAB1, we used a Duolink assay kit (Sigma) and followed the manufacturer’s instruction. In this assay, anti-rabbit-TAK1 (PA5-39743, ThermoFisher; 1:100 dilution) and anti-mouse-TXNIP (K0205-3, MBL; 1:200 dilution) or anti-mouse-TAB1 (sc-6053, SantaCruz; 1:50) were used. Briefly, permeabilized cells were incubated with blocking solution for 30 min at 37 °C. Cells were incubated with the rabbit TAK1 antibody and the mouse TXNIP Ab or mouse TAB1 Ab overnight at 4 °C and were washed and then were incubated with PLUS and MINUS PLA probes for 1 h at 37 °C. After washed 3 times, cells were incubated with the ligation mixture for 30 min at 37 °C. Washed 2 times and then were incubated with amplification mixture for 100 min at 37 °C. And were washed again with 1× wash buffer B. Cells were applied with Duolink mounting medium including DAPI. For confocal imaging, freshly isolated NK cells were plated, stimulated, fixed, and permeabilized as described above. After permeabilization, cells were blocked with PBS containing 5% BSA for 30 min at RT and stained primary antibodies overnight and secondary antibodies for 1 h at RT. Cells were mounted with Fluoroshield Mounting Medium with DAPI (Abcam). In this assay, p-c-Jun (3270, Cell signaling; 1:800 dilution), p-c-Fos (5248, Cell Signaling; 1:200 dilution), and p-TAK1 (PA5-39743, ThermoFisher; 1:100 dilution) were used. The images were captured with an LSM510 confocal microscope (Carl Zeiss, Gottingen, Germany).

### 4.9. Enzyme-Linked Immunosorbent Assay (ELISA)

IFN-γ, IL-1β, IL-6, and TNF-α level in serum and culture media were measured by Duo-Set antibody pairs (R&D Systems, IFN-γ; DY485, IL-1β; DY401, IL-6; DY406, TNF-α; DY410) following manufacturer’s instruction. Data were acquired by measuring at 450 nm on a microplate multi-reader (SpectraMax i3x; Molecular Devices).

### 4.10. Luciferase Reporter Assay

HEK 293 cells were transiently cotransfected with expression vectors, AP1–Luc plasmid (Stratagene, La Jolla, CA, USA) and cytomegalovirus-Renilla luciferase construct (pRL–CMV) (Promega, Madison, WI, USA), a Renilla-derived luciferase reporter plasmid for transfection control, using Lipofectamine and Plus reagent (Invitrogen, CA, USA) according to the manufacturer’s instructions. Luciferase assays were performed according to instructions supplied with the Luciferase Assay Kit (Promega, Madison, WI, USA). Firefly luciferase and Renilla luciferase activities were measured by the Dual Luciferase Reporter Assay System (Promega, Madison, WI, USA) on a luminometer (Turner Designs, Sunnyvale, CA, USA).

### 4.11. Quantitative Real-Time PCR

NK cells were isolated from the indicated mice, mRNA was extracted by using RNeasy Micro Kit (Qiagen) according to the manufacturer’s instructions. Total RNA (1 µg) was reverse-transcribed using cDNA synthesis kit (Toyobo, Osaka, Japan), and real-time PCR was performed in a Dice TP 800 Thermal Cyclear with SYBR Premix Ex Tag (Takara Bio, Kudatsu, Japan). All the measurements were normalized to the housekeeping genes Gapdh. We used the following primers: *Ifn-**γ*-F: 5′-AGCTCTTCCTCATGGCTGTT-3′, *Ifn-**γ*-R: 5′-TTTGCCAGTTCCTCCAGATA-3′; *Txnip*-F: 5′-TGACCTAATGGCACCAGTGT-3′, *Txnip*-R: 5′-GCCATTGGCAAGGTAAGTGT-3′; *Tlr2*-F: 5′-GGATAGGAGTTCGCAGGAGC-3′, *Tlr2*-R: 5′-TTGTTCCCTGTGTTGCTGGT-3′; *Gapdh*-F: 5′-CTGGCATTGCCCTCAACGAC-3′, *Gapdh*-R: 5′-CTTGCTGGGGCTGGTGGTCC-3′.

### 4.12. In Vitro Kinase Assay

Cell lysates (300 µg) were immunoprecipitated using anti-TAK1 antibodies (Cell Signaling Technology, Danvers, MA, USA). TAK1 immunoprecipitates were incubated with 2.5 µg of recombinant HIS tagged MKK6 protein (ATGen, Seongnam, Korea) and recombinant FLAG-tagged TXNIP protein (1 µg or 2 µg, Origene Technologies, Rockville, MD, USA) in 40 µL of 1× kinase buffer (Cell Signaling) containing 25 mM Tris-HCl (pH 7.5), 5 mM β-glycerophosphate, 2 mM dithiothreitol (DTT), 0.1 mM Na_3_VO_4_, 10 mM MgCl_2_, and 20 µM ATP at 30 °C for 20 min. The reactions were terminated by adding the SDS sample buffer and boiling for 5 min. Samples were then fractionated by 10% SDS-PAGE followed by Western blotting with antibodies to p-MKK3/6, MKK3/6, TXNIP, or TAK1 (Cell Signaling).

### 4.13. NK Cell Cytotoxicity Assay

NK cell-mediated cytotoxicity was examined using a standard 4 h ^51^Cr-release cytotoxicity assay. In brief, YAC-1 cells were incubated with 1.5 µCi of ^51^Cr for 1 h. The ^51^Cr-labeled YAC-1 cells (1 × 10^4^ cells per well) were placed into a 96 well round bottom plate in triplicate and co-cultured either with highly purified NK cells at the E:T ratios ranging from 40:1 to 10:1 for 4 h. The ^51^Cr-release from lysed YAC-1 cells by the NK cells was quantified using a γ-counter. The percentage of specific lysis was calculated according to the following formula: (experimental release—spontaneous release)/(maximum release - spontaneous release) × 100. Spontaneous release refers to ^51^Cr released from target cells in a complete medium alone, while maximum release refers to ^51^Cr released from target cells in a complete medium containing 1% Triton X-100. In order to investigate the effect of Pam3CSK4 stimulation on NK cell cytotoxicity between WT and KO NK cells, we pretreated NK cells with 1 µg/mL of Pam3CSK4 (Invivogen) for 16 h before cytotoxicity assay.

### 4.14. NK Cell Depletion and IFN-γ Neutralization

For in vivo depletion of NK cells, 200 µg of anti–NK1.1 antibody (PK136; purchased from BioXcell, Lebanon, NH, USA) or the same amount of Rat IgG control. The antibodies were intraperitoneally injected into the mice on Day-4 and Day-1 before Pam3CSK4 injection. For in vivo neutralization of IFN-γ, 500 µg of anti–IFN-γ antibody (XMG1.2; purchased from BioXcell) or the same amount of Rat IgG control. Antibodies were intraperitoneally injected into the mice at 17H before Pam3CSK4 injection.

### 4.15. NK Cells Adoptive Transfer, Pam3CSK4 Injection, and S. aureus Infection

*Rag2**^−/−^ × Il2rg^−/−^* C57BL/6 mice (*n* = 5) received highly purified NK cells (1 × 10^6^ cells; 99.5–99.9%, CD3^−^NK1.1^+^) from WT or KO mice intravenously. After 3 days, Pam3CSK4 (2 µg/g) was injected intraperitoneally into recipient mice. At 12 h after the injection, we sacrificed the recipient mice and analyzed cell populations (NK cells and Macrophages) in spleen and pro-inflammatory cytokines (IFN-γ, IL-1β, and IL-6) in serum and the activating receptors (CD80 and 86) in Mac-1^+^F4/80^mid^. *S.aureus* (4 × 10^7^ CFU), which is one of the representative Gram-positive bacteria, were injected into the recipient mice. At 12 h after the infection, we sacrificed the recipient mice and analyzed cell populations (NK cells and macrophages) in spleen, pro-inflamatory cytokines (IFN-γ, IL-1β, and IL-6) in serum and the activating receptors (CD80 and 86) in Mac-1^+^F4/80^mid^. For bacterial colony-forming unit (CFU) analysis, blood and tissue homogenates of kidney and blood from the experimental animals were serially diluted in phosphate-buffered saline and used to inoculate nutrient-medium plates. After incubation at 37 °C overnight, CFUs were counted and expressed as Log_10_ of CFU/mL of blood or Log_10_ of CFU/g of kidney.

### 4.16. Statistical Analysis

The data are expressed as the mean ± SD of *n* determinations and statistical significance was determined using Student’s *t*-tests. * *p* < 0.05, ** *p* < 0.005, *** *p* < 0.001, ns (not significant).

## Figures and Tables

**Figure 1 ijms-21-09499-f001:**
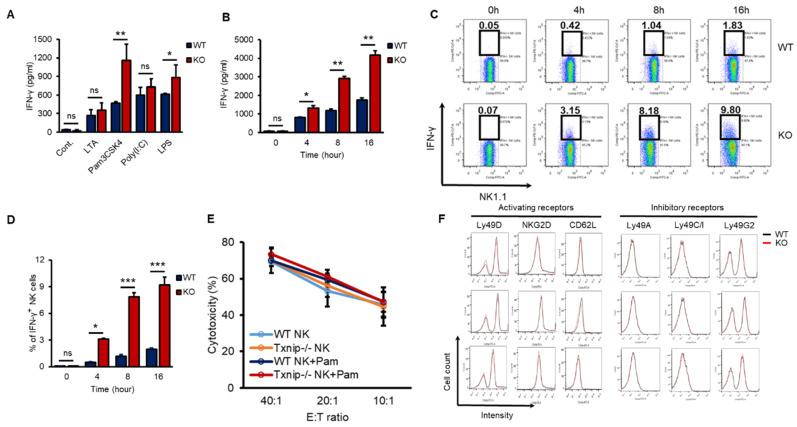
The loss of *Txnip* induces the production of IFN-γ in NK cells under various TLRs agonist treatment conditions. (**A**) Splenic NK cells from WT and KO mice were cultured at 1 × 10^6^ cells per well in 24-well plate and treated with LTA (1 µg/mL), Pam3CSK4 (Pam) (1 µg/mL), Poly(I:C) (1 µg/mL) and LPS (1 µg/mL) for 18 h (*n* = 3). Repeated three times. (**B**) WT and KO splenic NK cells were cultured at 1 × 10^6^ cells per well in 24-well plate and treated with Pam (1 µg/mL). Supernatants were harvested at indicated time point and IFN-γ concentration was determined using enzyme-linked immunosorbent assay (ELISA) (*n* = 3). Repeated three times. (**C**) Representative flow-cytometry plots. IFN-γ^+^ NK cells were stained intracellularly and analyzed by flow cytometry (*n* = 3). Repeated three times. (**D**) The frequency of IFN-γ^+^ NK cells (*n* = 3). Repeated three times. (**E**) Splenic NK cells were stimulated with Pam (1 µg/mL) for 16 h. The percent cytotoxicity from ^51^Cr release assay was shown for the NK cells stimulated by Pam3CSK4 isolated and cultured with radio-labeled YAC-1 target cells for 4 h at the indicated effector-to-target ratios (*n* = 3). Repeated three times. (**F**) The expression of activating (Ly49D, NKG2D, and CD62L) and inhibitory receptors (Ly49A, Ly49C/I, and Ly49G2) was analyzed by flow cytometry on WT and KO NK cells at indicated time after Pam3CSK4 treatment. Representative histogram profiles for each receptor expressed on WT and KO NK cells (*n* = 3). Repeated three times. Data are mean ± SD. Statistical significance was determined using Student’s *t-*tests. * *p* < 0.05, ** *p* < 0.01, *** *p* < 0.001, ns (not significant).

**Figure 2 ijms-21-09499-f002:**
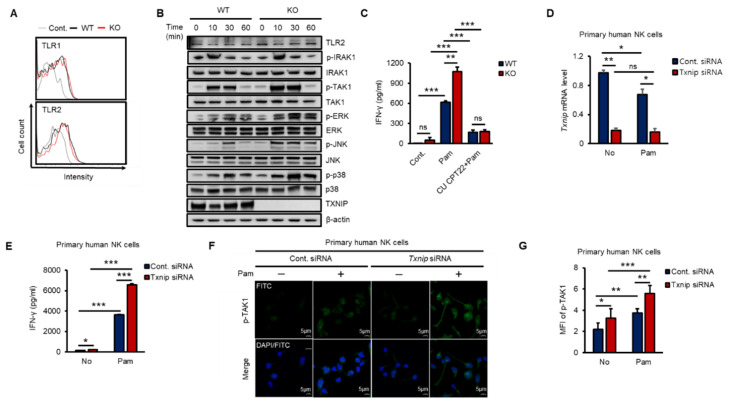
The loss of *Txnip* induces the activation of TAK1 and IFN-γ production in NK cells during Pam3CSK4 stimulation. (**A**) Representative images of FACS histograms. WT and KO NK cells were cultured at 5 × 10^5^ cells per well in 48-well plate and treated with Pam (1 µg/mL) for 1 h. WT NK cells; black line, KO NK cells; red line (*n* = 3). Repeated three times. (**B**) WT and KO NK cells were cultured at 1–2 × 10^6^ cells per well in 24-well plate and stimulated with Pam3CSK4 (1 µg/mL) for indicated time. The levels of TLR1/2 signaling molecules were analyzed by Western blotting. Repeated three times. (**C**) The production of IFN-γ was blocked by TLR1/2 signaling inhibitor, CU CPT22, in NK cells (*n* = 3). Repeated three times. (**D**) Knockdown of *Txnip* mRNA level in human primary NK cells was confirmed by quantitative real-time PCR at 42 h after electroporation (*n* = 3). Repeated three times. (**E**) Production of IFN-γ in human NK cells transfected with 300 nmol/100 µL of control or *Txnip* siRNA. NK cells were cultured at 1 × 10^6^ cells per well in 24-well plate and treated by Pam3CSK4 (1 µg/mL) for 18 h (*n* = 3). Repeated three times. (**F**) Representative confocal images for the phosphorylation of TAK1. Knockdown of *Txnip* induced the phosphorylation of TAK1 in human NK cells. NK cells were seeded on fibronectin-coated glass coverslips and treated with Pam (1 µg/mL) for 30 min. Repeated three times. (**G**) Measurement of mean fluorescent intensity (MFI) of phospho-TAK1 in human NK cells in (**F**) (*n* = 3). Repeated three times. Data are mean ± SD. Statistical significance was determined using Student’s *t-*tests. * *p* < 0.05, ** *p* < 0.01, *** *p* < 0.001, ns (not significant).

**Figure 3 ijms-21-09499-f003:**
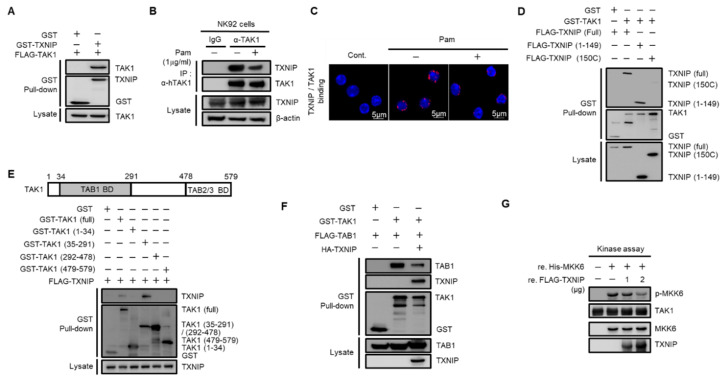
TXNIP directly interacts with TAK1 and inhibits TAK1 kinase activity. (**A**) TXNIP interacts with TAK1. GST fused proteins were pulled down with glutathione beads and then immunoblotted with α-FLAG and α-GST antibodies. (**B**) Endogenous binding between TAK1 and TXNIP following Pam (1 µg/mL) treatment for 1 h. NK92 cell lysates were immunoprecipitated with TAK1 antibody. (**C**) In situ PLA images for TXNIP-TAK1 binding in freshly isolated splenic NK cells. NK cells were treated with Pam (1 µg/mL) for 1 h. (**D**) TXNIP interacts with TAK1 via its 1–149 domain. (**E**) TAK1 interacts with TXNIP via its 35–291 domain. HEK 293 cells were transiently cotransfected each of the four domains (1–34, 35–291, 292–478, and 479–579) including full-length and TXNIP-encoding vectors. (**F**) TXNIP expression inhibits the TAK1-TAB1 complex formation. (**G**) In vitro kinase assay. TAK1 immunoprecipitates were incubated with 2.5 µg of MKK6 and TXNIP (1 µg or 2 µg) in 40 µL of 1× kinase buffer at 30 °C for 20 min. Western blotting was performed with antibodies to p-MKK3/6, MKK3/6, TXNIP, or TAK1. Results represent a representative data of three independent experiments.

**Figure 4 ijms-21-09499-f004:**
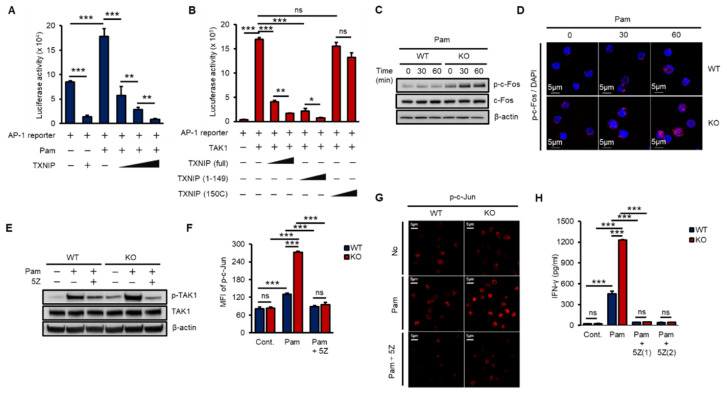
TXNIP regulates IFN-γ production in NK cells by inhibiting TAK1/AP-1 pathway. (**A**) TXNIP inhibits the transcriptional activity of AP-1 following Pam (1 µg/mL) treatment. AP-1 luciferase reporter and pRL-CMV vector were cotransfected with increasing amounts of FLAG-TXNIP into HEK 293 cells and then cells were stimulated with Pam (1 µg/mL) for 16 h (*n* = 3). Repeated two times. (**B**) The interaction between TXNIP and TAK1 is critical for the regulation of AP-1 transcriptional activity. AP-1luciferase reporter, pRL-CMV vector, and FLAG-TAK1 vector were cotransfected with increasing amounts of FLAG-TXNIP, FLAG-TXNIP (1–149) or FLAG-TXNIP (150C) into HEK 293 cells (*n* = 3). Repeated two times. (**C**) Induction of c-Fos phosphorylation in KO NK cells. WT and KO NK cells were cultured at 1 × 10^6^ cells per well in 24-well plate and stimulated with Pam3CSK4 as indicated time. Repeated three times. (**D**) Representative confocal images of phospho-c-Fos in NK cells following Pam treatment. Repeated three times. (**E**–**G**) TAK1 dependent activation of AP-1 and IFN-γ production in NK cells. WT and KO NK cells were cultured at 1 × 10^6^ cells per well in 24-well plate and stimulated by Pam for 1 h with 5z-7-oxozeaenol (5Z) (1 µM) or not. Repeated three times. (**E**) Western blot analysis show levels of phospho-TAK1. Repeated three times. (**F**) Phosphorylation of c-Jun was determined by flow cytometry (*n* = 3). Repeated three times. (**G**) The level of phospho-c-Jun was determined by confocal imaging with phospho-c-Jun antibody. Repeated three times. (**H**) Inhibition of IFN-γ production by TAK1 inhibitor, 5Z, in NK cells. WT and KO NK cells were stimulated by Pam (1 µg/mL) with or without 5Z (1 µM and 2 µM) for 16 h (*n* = 3). Repeated three times. Data are mean ± SD. Statistical significance was determined using Student’s *t-*tests. * *p* < 0.05, ** *p* < 0.01, *** *p* < 0.001, ns (not significant).

**Figure 5 ijms-21-09499-f005:**
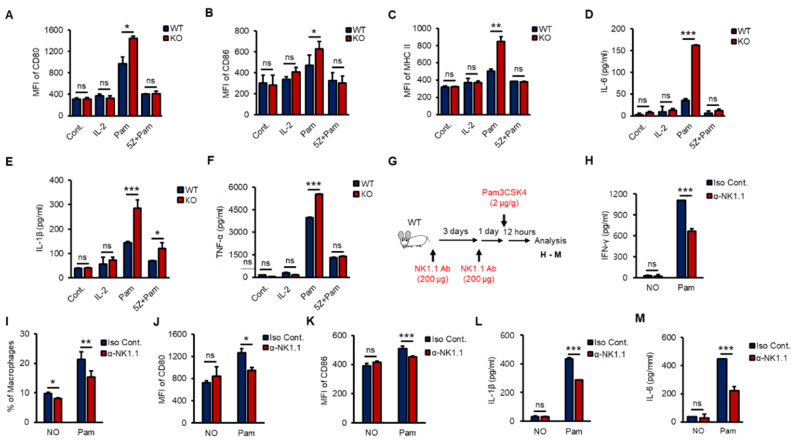
TAK1-dependent production of IFN-γ in vitro and NK cell-derived activation of macrophages in vivo. (**A**–**F**) WT and KO NK cells were pre-treated with 5Z (1 µM) for 1 h before Pam (1 µg/mL) stimulation for 12 h and then PBS washed NK cells were cocultured with RAW264.7 cells for 12 h. The expression of CD80 (**A**), CD86 (**B**), and MHC II (**C**) in RAW264.7 cells was determined by flow cytometry. The level of IL-6 (**D**), IL-1β (**E**), and TNF-α (**F**) was measured by ELISA assays in supernatants from the cocultured plates (*n* = 3). Repeated three times. (**G**) The procedure to examine the effects of NK cell depletion on the inflammation in vivo. (**H**) IFN-γ level in mice serum. (**I**) The frequency of macrophages in spleen. (**J**,**K**) The surface expression of CD80 (**J**) and CD86 (**K**) on Macrophages in spleen. (**L**,**M**) The concentration of IL-1β (**L**) and IL-6 (**M**) in mice serum (*n* = 5). Repeated three times. Data are mean ± SD. Statistical significance was determined using Student’s *t-*tests. * *p* < 0.05, ** *p* < 0.01, *** *p* < 0.001, ns (not significant).

**Figure 6 ijms-21-09499-f006:**
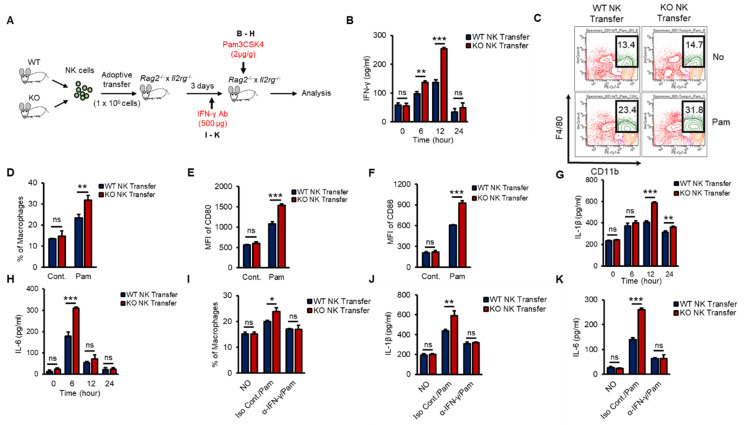
Differential induction of IFN-γ in KO NK cells results in the activation of macrophages in vivo during Pam stimulation. (**A**) Experimental design. WT or KO NK cells were adoptively transferred into *Rag2^−/−^ × Il2rg^−/−^* mice (1 × 10^6^ cells/mouse) intravenously (i.v). IFN-γ (500 µg/mouse) or Pam (2 µg/g) was intraperitoneally injected into recipient Rag2^−/−^ × Il2rg^−/−^ mice. (**B**) The concentration of IFN-γ in the serum from *Rag2*^−/−^ × *Il2rg*^−/−^ mice at the indicated time after Pam stimulation. (**C**,**D**) Frequency of macrophages (CD11b^+^F4/80^+^) in the spleen at 12 h after Pam stimulation. (**E**,**F**) The expression of CD80 (**E**) and CD86 (**F**) on the surface of macrophages in spleen at 12 h after Pam stimulation. (**G**,**H**) The concentration of IL-1β (**G**) and IL-6 (**H**) in the serum from *Rag^−/−^ × Il2rg^−/−^* mice received WT or KO NK cells at 0, 6, 12, 24 h after Pam stimulation. (**I**–**K**) IFN-γ neutralization mice model. Reduction of macrophage activation by IFN-γ neutralization in WT and KO NK cell-received *Rag2^−/−^ × Il2rg^−/−^* mice was determined by flow cytometry and ELISA assay. (**I**) Frequency of macrophages in spleen. The concentration of IL-1β (**J**) and IL-6 (**K**) in the serum. Experiments were independently repeated three times, and data are mean ± SD (*n* = 5). Statistical significance was determined using Student’s *t-*tests. *n* = 5, * *p* < 0.05, ** *p* < 0.01, *** *p* < 0.001, ns (not significant).

**Figure 7 ijms-21-09499-f007:**
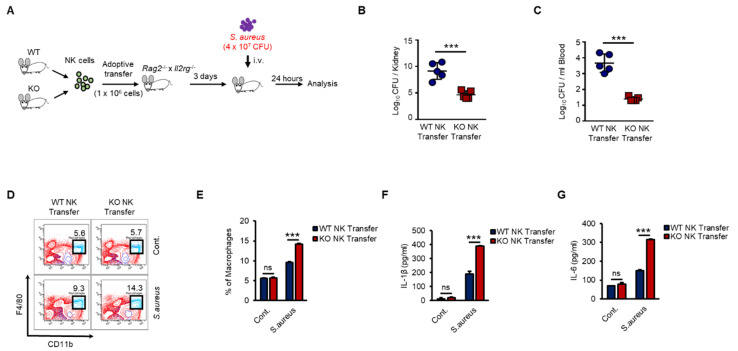
KO NK cells highly induces the activation of macrophages and the bacterial clearance than WT NK cells under *S.aureus* infection in vivo. (**A**) Experimental design. WT and KO NK cells were adoptive transferred i.v. into *Rag^−/−^ × Il2rg^−/−^* mice. Three days later, recipient mice were infected i.v. with *S.aureus* (4 × 10^7^ CFU) and sacrificed after 24 h. (**B**,**C**) Bacterial loads in the kidneys (**B**) and blood (**C**) of WT or KO NK cell-received mice. Each symbol represents the bacterial counts determined in an individual mouse and the horizontal lines represent the average ± SD for each mouse. (**D**) Representative dot plots of macrophages (CD11b^+^F4/80^+^) in spleen. (**E**) Frequency of macrophages (CD11b^+^F4/80+) in spleen. (**F**,**G**) IL-1β level (**F**) and IL-6 level (**G**) in the serum of WT or KO NK cell-received mice. Experiments were independently repeated three times, and data are mean ± SD (*n* = 5). Statistical significance was determined using Student’s *t-*tests. *n* = 5, *** *p* < 0.001, ns (not significant).

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
