# Peer review of "TXNIP Regulates Natural Killer Cell-Mediated Innate Immunity by Inhibiting IFN-γ Production during Bacterial Infection"

_ijms, 2020, doi:10.3390/ijms21249499_

Round 1
Reviewer 1 Report
This manuscript provides a very thorough investigation of the IFNg regulation by TXNIP in NK cells. The authors demonstrated that TXNIP directly interacted with TAK1 and thus inhibiting IFN-γ production in NK cells during Pam3CSK4 stimulation Staphylococcus aureus (S. aureus) infection. It is well written in English and the data is well presented. I will recommend acceptance after minor revision.
Minor comment:
- It is interesting to see TXNIP is mainly involved in Pam stimulation, as well as a small increase in LPS stimulation as shown in Figure 1A. Could the authors discuss the potential mechanism of such preference of TXNIP.
- For figure 1F, there seems no phenotype difference between WT and KO of TXNIP. Did the author try RNAseq or other more robust method to profile any phenotype difference?
- Is there any evidence that TXNIP could play this similar function in other cell types, such as DC, macrophage and B cells where TLR pathway is also a critical pathway.
Author Response
We appreciate the reviewer’s comprehensive comments throughout the manuscript. Please check attached file.

Reviewer 2 Report
In this study, Kim et al. provide evidence for a role of Txnip in regulation of IFN-g production in NK cells. The authors show that Txnip was able to interfere with the formation of the TAK1/TAB1 complex and to inhibit the production of IFN-g in NK cells. Moreover, Txnip deletion in NK cells led to an improved IFN-g-dependent host protection, by promoting activation of macrophages.
The manuscript is interesting and novel, however, some key aspects need to be addressed.
- In the paper published in 2005 (Lee at al., Immunity), the authors provided evidence for a role of Txnip in regulation of NK cell development. In that study, Txnip-/- mice showed a limited number of NK cells, and this aspect is not considered, at all, in the current work. Thus, this study would highly benefit from a better profiling of the NK cell subsets left in the Txnip-/- mice (CD27, CD11b, KLRG1, CD49b, CD49a, etc.). In this context, it is not clear how transferred Txnip-/- NK cells can survive, expand, and function in vivo, in relation to the impact of Txnip deletion observed in the previous Immunity paper (2005). This aspect is interesting and perplexing at the same time, and it has to be addressed. Frequency of NK cells after adoptive transfer needs to be shown.
- Some of the flow cytometry data require the appropriate controls, in particular, Figure 1F; perforin staining (NK cells should be highly positive for perforin); TLR1/2 staining (not convincing). Moreover, what is the percentage of IFN-gamma-producing NK cells after treatment with siRNA for Txnip (Figure 2F)?
- As stated by the authors, cells are kept and stimulated in presence of IL-15 in all the experiments. Is Txnip involved in regulation of IL-15 signaling in NK cells? In other words, could the survival and ability to produce IFN-gamma be regulated by a differential sensitivity to IL-15 in Txnip-/- NK cells? Moreover, how is the ability of NK cells to produce IFN-g following treatment with cytokines, such as IL-12 and IL-18?
Author Response

(The authors gave the same response as above.)

Reviewer 3 Report
This reviewer has no major comments outside of an admiration of the excellent work done within. I summarize experiments as I interpreted them and point out a couple of things that could mildly improve this already impressive manuscript.
In Figure 1, the authors first show that loss of Txnip induces IFN-γ production in NK cells over time under a wide variety of stimulatory conditions, although this effect is greatest with Pam stimulation. Thus, based on this result the authors focused on the TLR1/TLR2 signaling pathway. Elevated IFN-γ secretion and production by TXNIP KO cells was shown using ELISA and flow cytometry, respectively. No differences were noted in the expression of activating or inhibitory receptors on the KO or WT NK cells. Cytotoxic efficacy of WT and TXNIP KO NK cells against YAC-1 cells was not significantly different indicating that although IFN-γ production was differentially modulated, cytotoxicity was unaltered with the Pam stimulant (also no differences in TNF-α or perforin were noted). These experiments are indeed convincing that TXNIP regulates IFN-γ production in NK cells via TXNIP.
The authors next sought to determine the mechanism by which TXNIP modulates IFN-γ production in the TLR1/2 mediated innate signaling pathway. Pam stimulation did not alter the expression of either TLR 1 or 2 following 12 hours of Pam treatment. Since the Toll-like receptors in WT and KO NK cells did not appear to be differentially expressed following Pam treatment, the authors examined the activation status of multiple down-stream effector protein via phospho-specific Western blotting. A very modest increase in p-TAK1 was observed (this really needs densitometric analysis to be meaningful…) and modest increases in (p-) ERK, JNK, and p38 were observed. The data with p-JNK is the most convincing at low and high conc. of Pam. Supplementary data show that in WT NK cells, TXNIP is upregulated while IFN-γ is downregulated, demonstrating a direct link in WT NK cells for TXNIP in the negative regulation of IFN-γ production. Indeed, siRNA knockdown of TXNIP in WT NK cells was shown to downregulate IFN-γ production in primary human NK cells. This reviewer appreciates this experiment as it adds a significant level of physiological relevance to human mechanisms of immunity. Combined with earlier observations, the authors conclude that TXNIP must normally interact with TAK1 to downregulate or inhibit IFN-γ production in the TLR1/2 pathway. These data are overall convincing but the central piece of data here is the Western blots, which need some sort of quantitation and preferably multiple samples with statistical analysis.
With their targets identified, the authors proceed to dissect out the mechanistic interactions between TAK1 and TXNIP. The authors were able to immunoprecipitated TAK1 with TXNIP in HEK cells when ectopically expressed. Using a series of TXNIP and TAK1 deletion mutants, the authors determined thatTXNIP interacts wit the TAB1 binding region of TAK1. This is an important mechanistic finding becauseTAB1 phosphorylates factors that regulate JNK and p38 activation/phosphorylation. These data are in line with the Western blotting data shown in figure 2. Indeed, TXNIP appeared to specifically interfere with TAB1 but not 2/3. Pam treatment increased the interaction between TAK1 and TAB1 while the absence of TXNIP significantly increased this interaction. Furthermore, recombinant TXNIP did indeed inhibit the kinase activity TAK1, notably, in a dose-dependent manner suggesting a 1:1 interaction. Thus, the authors convincingly show that TXNIP binds TAK1, interfering with its ability to interact with TAB1 and thus active kinase capability.
The authors next sought to determine whether modulation of TAK1 influenced key transcription factors known to regulate IFN-γ production. AP-1 is the major transcriptional regulator of IFN-γ in this pathway and thus the authors investigated whether AP-1 is modulated in a TXNIP-TAK-1 dependent manner. The authors used an AP-1 luciferase reporter assay to assess AP-1 activation in response to Pam treatment with varying amounts of TXNIP. Notably, there was a dramatic effect of increased TXNIP on the suppression of AP-1 activation. Using this same assay, the authors demonstrated that the TAK1 binding domain of TXNIP is essential for suppression of AP-1 activation following Pam stimulation. Phosphorylation of c-Fos was significantly elevated in TXNIP KO NK cells following Pam stimulation. Using the Tak1 inhibitor 5Z, the authors demonstrated that inhibition of TAK1 does indeed reduce c-Jun phosphorylation and IFN-γ production. These results convincingly demonstrate that during stimulation, TXNIP interferes with TAK1 mediated activation of AP-1 and downstream IFN-γ production.
The authors next sought to link TXNIP and NK cell mediated activation of macrophages in vivo. Indeed, the absence of TXNIP appears to increase macrophage activation and cytokine production with Pam stimulation. Furthermore, this effect requires TAK1.
In an elegant series of experiments that authors next sought to determine whether the autonomous function of TXNIP in NK cells impacted macrophage activation. The authors used Rag KO mice and adoptively transferred NK cells from TXNIP KO mice and examined IFN-γ production, other cytokines, and macrophage activation. Indeed, in line with previous experiments, the absence of TXNIP appears to upregulate macrophage activation and inflammatory cytokines. To determine whether this effect was the direct result of increased IFN-γ produced by TXNIP KO NK cells, the authors used a neutralizing antibody against IFNγ. In line with the expectations of the great work done thus far, blocking IFN-γ effectively ameliorated effects in the TXNIP KO, thus demonstrating that increased IFN-γ produced in the absence of TXNIP is directly responsible for elevated pro inflammatory cytokine production and increased macrophage activation. The data demonstrate that TXNIP plays a direct and autonomous role in modulating macrophage activation and IFN-γ production.
In a final set of experiments that add an important level of physiological relevance with respect to the observations thus far, the authors examined the ability of TXNIP KO NK cells to clear a Staph. infection. The authors again used their adaptive transfer system with Rag KO mice receiving TXNIP KO NK cells. Strikingly, mice receiving TXNIP KO NK cells showed dramatically reduced bacterial loads which correlated with increased inflammatory cytokine production.
This reviewer is extremely impressed with the high caliber of scientific work done in this study.
Author Response

(The authors gave the same response as above.)

Round 2
Reviewer 2 Report
Since the authors did not provide convincing data regarding my point 1 (see below), these aspects have to be carefully discussed in the discussion session, as a limitation of the study.
In the paper published in 2005 (Lee at al., Immunity), the authors provided evidence for a role of Txnip in regulation of NK cell development. In that study, Txnip-/- mice showed a limited number of NK cells, and this aspect is not considered, at all, in the current work. Thus, this study would highly benefit from a better profiling of the NK cell subsets left in the Txnip-/- mice (CD27, CD11b, KLRG1, CD49b, CD49a, etc.). In this context, it is not clear how transferred Txnip-/- NK cells can survive, expand, and function in vivo, in relation to the impact of Txnip deletion observed in the previous Immunity paper (2005). This aspect is interesting and perplexing at the same time, and it has to be addressed.
